# Recipients of Suicide-Related Disclosure: The Link between Disclosure and Posttraumatic Growth for Suicide Attempt Survivors

**DOI:** 10.3390/ijerph16203815

**Published:** 2019-10-10

**Authors:** Laura M. Frey, Christopher W. Drapeau, Anthony Fulginiti, Nathalie Oexle, Dese’Rae L. Stage, Lindsay Sheehan, Julie Cerel, Melinda Moore

**Affiliations:** 1Couple & Family Therapy Program, Kent School of Social Work, 136A Burhans Hall, Shelby Campus, University of Louisville, Louisville, KY 40292, USA; 2Department of Education, Valparaiso University, Valparaiso, IN 46383, USA; 3Graduate School of Social Work, University of Denver, Denver, CO 80208, USA; 4Department of Psychiatry II, Ulm University and BKH Günzburg, Baden-Württemberg, 89081 Ulm, Germany; 5Live Through This, Philadelphia, PA, USA; 6Lewis College of Human Sciences, Illinois Institute of Technology, Chicago, IL 60616, USA; 7College of social Work, University of Kentucky, Lexington, KY 40506, USA; 8Department of Psychology, Eastern Kentucky University, Richmond, KY 40475, USA

**Keywords:** family dynamics, lived experience, posttraumatic growth, suicide disclosure, suicide prevention

## Abstract

It is important to explore factors that could help or hinder one’s wellbeing following a suicide attempt, which could yield not only negative consequences but also posttraumatic growth (PTG; positive changes following a traumatic event). The present study used a multivariate analysis of covariance (MANCOVA) to test the relationship between disclosure, PTG, and posttraumatic depreciation among suicide attempt survivors when controlling for time since attempt and to test whether these effects remained after controlling for quality of support from family and friends. Suicide attempt survivors (*n* = 159) completed an online survey about their experiences. Increases in disclosure to family and friends but not to healthcare providers predicted changes in PTG. The effects of family disclosure remained even after controlling for quality of support. Disclosure to healthcare providers demonstrated some statistical effects on PTG, yet in the opposite direction and only after controlling for quality of support. The control variables—time since attempt and quality of support—were the only variables that predicted a change in posttraumatic depreciation. These findings suggest there is value in disclosing one’s personal story to family regardless of whether one receives supportive responses and that social support can impact one’s PTG.

## 1. Introduction

Over the past decade, the field of Suicidology has witnessed a lived-experience movement that aims to increase inclusion for suicide attempt survivors in advocacy, treatment, and research. *Lived experience* is a term used to describe individuals who have had personal experiences with suicidal behavior. This term is purposefully broad, so that it may include those who have experienced their own suicidal experiences (e.g., those who have experienced suicidal ideation, and attempt survivors) and those who have experienced someone else’s suicidal experiences (e.g., those who have lost a loved one to suicide. The efforts to increase inclusion included a new membership division started in 2014 for attempt survivors in the American Association of Suicidology as well as recent reports providing guidance for the field that were written collaboratively by attempt survivors and professionals [1,2]. In the past, much of the research on attempt survivors focused on the negative ramifications of an attempt, which made sense given the despair leading up to a suicide attempt [3], the risk that occurs post-attempt [4], and the stigma that occurs as a result [5]. However, with the emergence and strengthening of the lived experience movement, attention has shifted towards exploring recovery post-attempt. 

Although surviving a suicide attempt can be a traumatic event, posttraumatic growth (PTG)—increased levels of personal growth in goals, behaviors, beliefs, and/or identity following a crisis or traumatic experience [6,7]—can also occur. Emerging research has applied this concept to suicide loss survivors [8,9]. Stories of positive insights as a result of the loss emerged during interviews with those bereaved by suicide, but loss survivors were commonly hesitant to admit this outcome to others [10]. A recent study attempted to highlight the mechanisms through which PTG in loss survivors occurs, linking increased PTG to more time since the death, a perceived closeness with the deceased, and social support [11]. PTG scholars have also explored the role of posttraumatic depreciation, which refers to negative consequences of a traumatic experience [12,13]. Rather than viewing growth and depreciation as two bipolar constructs, scholars have posited that individuals often experience both gains and losses simultaneously following a traumatic event. Overall, studies linking both PTG and posttraumatic depreciation warrant exploring both concepts for attempt survivors. 

Disclosure about one’s traumatic experience is a mechanism that has been shown to promote posttraumatic growth. For example, this link has been found among suicide loss survivors [14] as well as other kinds of survivors, such as survivors of accidental injury [15] and sexual victimization [16]. Although not tested directly on PTG outcomes, past research on suicide-related disclosure—the intentional sharing of one’s own current or past suicidal experiences to either seek help or share one’s personal history [17]—has also demonstrated positive outcomes. Disclosure when coupled with helpful reactions has been linked to less severe depression symptoms [18], as well as lower levels of burdensomeness and higher levels of belongingness [19]. A similar effect from disclosure and subsequent helpful reactions has been seen for PTG, in that helpful responses to disclosure were linked to PTG in survivors of traumatic brain injury [20]. Among loss survivors, not being able to disclose (i.e., secrecy) was a mediator between stigma and personal growth [21], suggesting that disclosure could be a key variable in posttraumatic growth. However, disclosure about one’s suicidal experiences can yield not only benefits, such as help-seeking and recovery, but also negative consequences, such as discomfort, unwanted treatment, and stigmatizing responses [22]. Therefore, it could be that disclosure yields both growth and depreciation following a suicide attempt.

Choosing to whom and how much to disclose are two aspects that might play a role in consequences of suicide disclosure. For example, past research has found that youth and young adults typically disclose to peers before family members (i.e., family of origin, also known as immediate family, or the members by and with whom one is raised) or professionals [23,24,25]. Studies on adult disclosure are inconsistent, with some studies reporting higher rates of disclosure to family over friends [26] as well as some friends over family [27,28] and professionals [28]. Our past work found a connection between type of recipient and stigma as a result of disclosure [18], suggesting that not all groups are likely to react in the same way. More specifically, stigma experienced following disclosure to social network members (e.g., family, friends, coworkers) was a better predictor of depression symptoms than stigma experienced from mental health professionals and nonmental health professionals. These works suggest that recipients of disclosure might play a role in PTG outcomes. Additionally, although not much research has examined how various levels of disclosure affect outcomes, our previous work indicated that a greater depth of disclosure (i.e., disclosing more information about the experience) was linked to more positive family reactions [19]. Similarly, past research has shown that greater levels of self-disclosure in general are linked to less severe suicidal behavior [29]. Combined, this research suggests a possible link between depth of suicide disclosure and PTG outcomes. 

The present study tested the relationship between suicide-related disclosure, posttraumatic growth, and posttraumatic depreciation among suicide attempt survivors. We hypothesized that a greater depth of disclosure is linked to higher levels of posttraumatic growth and lower levels of posttraumatic depreciation. Additionally, we measured depth of disclosure by recipient separately to explore whether a greater depth of disclosure to various recipients (i.e., one’s specific confidant, and family, friends, and healthcare professionals in general) demonstrated different effects on PTG outcomes. Finally, we tested whether these effects remained even after controlling for quality of support from family and friends. 

## 2. Method

### 2.1. Recruitment Strategy and Sample Characteristics

Study procedures were approved by the Institutional Review Board at Valparaiso University in the United States. Suicide attempt survivors were invited to participate in an online survey between October 2017 and June 2018. Invitations to participate were posted on email listservs of prominent suicide prevention organizations in the United States, such as the American Association of Suicidology, and on social media boards for attempts survivor support groups in the United States. The invitation stated participants would be asked about their experiences following their most recent suicide attempt and then stated, “We currently do not know much about whether suicide attempt survivors experience positive outcomes after a suicide attempt. It is our hope that your participation in this study will help us understand the experiences of attempt survivors in general, as well as the ways in which your interactions with others have affected your life following you most recent suicide attempt.”. To participate, respondents must have been at least 18 years old and self-identify as a suicide attempt survivor. Invitations included a link to an online survey hosted on Qualtrics. Participants provided consent by clicking “I agree” after reading a short description of the study aims, risk, and benefits, and participants were allowed to participate only one time. No monetary compensation was provided. 

The present study included a sample of 159 participants with ages ranging from 18–67 (*M* = 35.72, *SD* = 12.32). Surveys were initially begun by 304 respondents; however, some respondents were omitted for not completing the survey (*n* = 124) or for not completing items needed for key variables used in our analyses (*n* = 21). The majority of the sample was female-identified (88.1%) and non-Hispanic White (89.9%). Roughly 10% of the sample was male-identified, and 1.3% identified as transgender. Over half were heterosexual (59.1%), while 8.8% were lesbian or gay, 23.9% were bisexual, 5.7% listed sexuality as “other”, and 3.8% responded “don’t know”. Roughly 3% were Asian or Pacific Islander, 3.1% were Latinx or Hispanic, 0.6% were non-Hispanic Black, 0.6% were Native American or Aleut, and 2.5% identified as Other. Nearly half (47.8%) were single and had never married, 27% were currently married, 20.8% were divorced, 3.8% were separated, and 0.6% were widowed. Education levels varied: 5.7% had a high school diploma or equivalent, 9.4% had some college but less than a year, 28.3% had at least a year of college but no degree, 10.7% had an associate degree, 20.1% had a bachelor’s degree, 22.0% had a master’s degree, and 3.8% had a professional degree (e.g., MD, JD) or doctorate. Over half of the sample (56.6%) was currently employed for wages or self-employed, 5.0% worked in the home for no wages, 12.6% were students, 3.1% were retired, and 22.6% were unemployed (i.e., unable to work, looking for work, or out of work but not currently looking). Number of attempts ranged from 1 to 50 (*M* = 3.40, Median = 2.00; *SD* = 4.72). Nearly all (93.6%) participants had sought help through counseling, 60.7% through a physician, and 25.5% through a religious-oriented source. The majority (83.7%) had also sought help from a friend, 56.8% from a romantic partner, and 48.6% from a parent. A small percentage (3.8%) were currently attending an attempt survivor support group, 11.3% were not currently attending but had attended in the past, 49.7% had never attended, and 35.2% said they had never attended an attempt survivor support group and said they had not known that type of group existed. 

### 2.2. Measures

#### 2.2.1. Depth of Disclosure

Depth of disclosure was measured using two scales. First, the confidant depth of disclosure scale (CDDS) was developed for this study and used to assess how much a respondent disclosed about their current and past suicidal experiences to an identified emotional confidant, which we defined as the person with whom the respondent talks the most about their suicidal experiences. This scale is based on the self-harm and suicide disclosure scale [30]. First, respondents are asked to select a family member or friend with whom they most often share when they are feeling suicidal. Respondents are then presented with a list of 10 topics related to suicidal experiences and asked to indicate how often they share each specific type of information with that specific person on a scale from never (1) to almost always (4). Instructions direct respondents who have not experienced the type of information (e.g., for those who have only experienced ideation and not an attempt) to select not applicable. This measure is designed to assess disclosure to a specific confidant in contrast to disclosure that generally occurs with most people. Responses are then summed and divided by the number of applicable items to create a mean score ranging from 1 to 4. Cronbach’s alpha for this sample was 0.93.

Respondents also completed the global depth of disclosure scale (GDDS), a measure that expands on the CDDS to assess the degree to which respondents disclose their past suicidal experiences in general to family, friends, and healthcare providers. Respondents are presented with the same topics listed in the CDDS and asked to indicate how often they share the topic with each group of people using four response options ranging from *never* (1) to *almost always* (4). Instructions direct respondents who have not experienced the type of information (e.g., for those who have only experienced ideation and not an attempt) to select *not applicable*. Responses to the GDDS are summed for each group and averaged by the number of applicable items, resulting in a mean score ranging from 1 to 4 to indicate how much a respondent discloses to each of the three groups. Cronbach’s alpha for this sample was 0.91 for family, 0.93 for friends, and 0.95 for healthcare providers.

#### 2.2.2. Posttraumatic Growth and Depreciation

The paired-format posttraumatic growth inventory (PTGI-42) [12] was used to measure both growth and depreciation post-suicide attempt. This instrument is a variation of the posttraumatic growth inventory (PTGI) [31]. The PTG-42 presents respondents with 21 statements from the original PTGI and pairs them with the antonyms or alternatives to these items (e.g., “I find it difficult to clarify priorities about what is important in life” and “I have a diminished feeling of self-reliance”) [12]. Respondents are asked to indicate the degree to which each statement fits their experience using six response options ranging from I did not experience this change (0) to I experienced this change to a very great degree (5). Presenting these statements as pairs encourages participants both types of posttraumatic changes simultaneously. Mean scores with all positive and negative items separately were used to measure overall PTG and depreciation. We also calculated mean scores on five subscales [31,32,33] for PTG: (a) relating to others, which refers to having compassion for others (e.g., “I have a greater sense of closeness with others”); (b) new possibilities, which refers to behaving in new ways (e.g., “I am more likely to try to change things which need changing”); (c) personal strength, which refers to discovering that one is stronger than expected (e.g., “I have a greater feeling of self-reliance”); (d) spiritual change, which refers to developing a better understanding of spiritual matters (e.g., “I have a stronger religious faith”); and (e) appreciation for life, which refers to increased appreciation for the value of one’s own life (e.g., “I changed my priorities about what is important in life”). These scales demonstrated good internal reliability within our sample, with Cronbach’s alphas of 0.94 for overall depreciation., 0.96 for the total PTG scale, 0.91 for relating to others, 0.86 for new possibilities, 0.89 for personal strength, 0.85 for spiritual change, and 0.81 for appreciation of life. 

#### 2.2.3. Quality of Support

The 12-item multidimensional scale of perceived social support [34] was used to measure quality of support from family and friends. Respondents are asked to indicate the degree to which they agree with each statement (e.g., “My family really tries to help me”; “I can talk about my problems with my friends”.) using seven response options ranging from strongly disagree (1) to strongly agree (7). The scale includes three 4-item subscales indicating support from family, friends, and significant others. Only the family and friend subscales were used in the present study. Cronbach’s alpha demonstrated strong internal reliability for both the family (0.92) and friend (0.93) subscales.

### 2.3. Data Analysis

An a priori power analysis for a MANCOVA using G*Power [35] based on an alpha (α) of 0.05, a beta (β) of 0.20, a medium effect size (*f*^2^) of 0.06, 4 predictors, 3 covariates predictors, and 7 outcome variables yielded a recommended sample size of 238. However, a sample size of only 159 was available, which provided sufficient power to detect a medium-to-large [36] effect size of *f*^2^ = 0.09 and Pillai–Bartlett trace *V* = 0.06 and larger with an alpha (α) value of 0.05 and a beta (β) of 0.20. Given the homogeneity of the sample, we conducted preliminary analyses to determine whether participant demographics affected the disclosure variables, PTG, and posttraumatic depreciation. Independent *t*-tests were used to assess the effects of sex (male vs. female), race and ethnicity (non-Hispanic White vs. non-White), and gender and sexual diversity (cisgender and heterosexual vs. LGBTQ+). Hypotheses regarding the effects of disclosure depth to various groups on both PTG scales and depreciation were tested using a multivariate analysis of covariance (MANCOVA). We first entered confidant disclosure, family disclosure, friend disclosure, and healthcare disclosure as predictor variables, and the PTG scales and depreciation scale were tested as dependent variables while controlling for time since attempt. Then, we ran the same MANCOVA while also controlling for quality of support from both family and friends.

## 3. Results

### 3.1. Preliminary Analyses

No statistical differences were observed on main variables based on sex, racial and ethnic diversity, or gender and sexual diversity. Although not statistically significant at the 0.05 level, the results suggest females, on average, disclosed more to family compared to males, *t*(140) = −1.94, *p* = 0.054, *M_Difference_* = 0.41. Given that none of these variables were statistically significant, they were not included as control variables in the final analyses. Table 1 displays the means, standard deviations, and intercorrelations for independent and dependent variables. All four disclosure measures were statistically and positively correlated with moderate-to-large effect sizes [36]. Overall PTG was correlated with all disclosure variables except healthcare disclosure, indicating small-to-medium effect sizes. Posttraumatic depreciation demonstrated a medium statistical correlation with PTG, but it was not correlated statistically with any of the disclosure variables. Time since attempt was moderately correlated with posttraumatic depreciation, linking respondents whose most recent attempt occurred within the past five years with higher levels of depreciation. Finally, family support was positively and statistically correlated with disclosure to a confidant and family, while friend support was positively and statistically correlated with disclosure to friends and healthcare providers. 

### 3.2. Main Analyses

Next, a MANCOVA was conducted to examine whether disclosure depth to a confidant, family, friends, and healthcare providers predicted changes in various types of PTG and depreciation when controlling for time since attempt. Table 2 and Table 3 display findings from multivariate and univariate tests, respectively. Using Pillai’s trace from multivariate tests, there were statistical effects of family disclosure, *V* = 0.16, *F*(4,155) = 2.56, *p* = 0.019, and friend disclosure, *V* = 0.21, *F*(4,155) = 3.59, *p* = 0.002, on PTG and depreciation. Multivariate tests did *not* indicate statistical effects for confidant, healthcare disclosure, or time since attempt. 

Univariate analyses demonstrated interesting direct effects for both family and friend disclosure (see Table 3). Family disclosure had a statistical effect on overall PTG (*B* = 0.58, *p* = 0.001), as well as PTG subscales for relating to others (*B* = 0.58, *p* = 0.002), new possibilities (*B* = 0.65, *p* = 0.001), personal strengths (*B* = 0.66, *p* = 0.003), and appreciation of life (*B* = 0.66, *p* = 0.002). Parameter estimates indicated that increases in depth of disclosure to family predicted increases on each of these PTG scales. The only subscale to not have a statistical effect from family disclosure was spiritual change. Disclosure to friends had a statistical effect on overall PTG (*B* = 0.47, *p* = 0.009), as well as PTG subscales for relating to others (*B* = 0.55, *p* = 0.004), personal strengths (*B* = 0.47, *p* = 0.030), and spiritual change (*B* = 0.82, *p* = 0.001). Similar to family disclosure, parameter estimates indicated that depth of disclosure to friends had a positive statistical effect for each of these subscales. Healthcare disclosure had statistical effects on PTG subscales for appreciation of life only, and parameter estimates indicated these were negative effects, with increases in healthcare disclosure predicting decreases in appreciation of life (*B* = −0.34, *p* = 0.048). Time since attempt had a statistical effect on the PTG subscale for new possibilities (*B* = 0.54, *p* = 0.036), with respondents whose most recent attempt occurred more than five years ago reporting higher rates of developing new interests post-attempt compared to those who attempted within the past five years. Time since attempt was the only variable to predict a statistical change in posttraumatic depreciation (*B* = −0.49, *p* = 0.012), indicating lower rates of posttraumatic depreciation for those who attempted suicide more than five years ago. Finally, it is important to note that degree of disclosure to one’s confidant did not display statistical effects for any of the PTG scales or depreciation or overall scores.

A similar MANCOVA was performed with perceived family and friend support included as additional covariates. Family and friend support demonstrated statistical effects on both PTG and depreciation overall as well as various PTG subscales, and quality of support accounted for the greatest amounts of variance in outcome variables compared to other predictor variables (based on partial eta-squared [η^2^] values). Even after controlling for perceived family and friend support, family disclosure demonstrated the same effects on PTG scales. Friend disclosure showed fewer effects, with only its effect on spiritual change remaining statistically significant. By contrast, some effects of healthcare disclosure emerged: Healthcare disclosure had statistical effects on PTG overall (*B* = −0.27, *p* = 0.039) and the PTG subscales of new possibilities (*B* = −0.30, *p* = 0.049) and appreciation for life (*B* = −0.40, *p* = 0.014). It is important to note that these effects were reversed compared to other disclosure effects, with more disclosure to healthcare providers predicting lower levels of PTG. Furthermore, the statistical direct effects of time since attempt disappeared after controlling for quality of support.

## 4. Discussion

The present study tested whether various types of disclosure (i.e., disclosure to a confidant, family, friends, and healthcare providers) predicted changes in posttraumatic growth (PTG) and depreciation for suicide attempt survivors when controlling for time since attempt. We also tested whether these effects remained after controlling for the quality of family and friend support. Increases in disclosure to family and friends were the only types to predict changes in PTG. The effects of family disclosure also remained even after controlling for quality of support, yet most of the statistical effects from disclosure to friends disappeared. Disclosure to healthcare providers demonstrated some statistical effects on PTG, yet in the opposite direction and only after controlling for quality of support. The control variables—time since attempt, and then quality of support but not time since attempt—were the only variables that predicted a change in posttraumatic depreciation.

An important takeaway from the present study is that disclosure to family members can promote posttraumatic growth (PTG). More specifically, greater depth of disclosure was statistically associated with greater PTG, even when accounting for perceived family social support. Finding that family is involved in the PTG process after a suicide attempt is not entirely unexpected. Others have already made a compelling case for family as a prominent, albeit underappreciated, force in the development of suicidal experiences [37]. Our results add to the literature by moving beyond a risk orientation with respect to the influence of family in suicide events and toward a recovery orientation that seeks to understand the contribution of family in the emergence of positive change that could occur following a suicide attempt. Moreover, these results point to the fact that disclosure to family can be a vehicle for positive posttraumatic change independent of the support provided by them, which has also been seen in other stigmatized populations (e.g., when disclosing HIV status to intimate partners; [38]). Aside from support, it is difficult to know *why* disclosure to family can make a difference in PTG. It might have something to do with the unique nature of familial relationships. Although “family” can mean different things to different people, family-of-origin ties are special in that they are ascribed—people do not choose their family members. Being able to openly share a life-changing event, such as surviving a suicide attempt, with one’s family of origin may carry special significance. For example, it could be cathartic to have shared that an identifying moment occurred regardless of whether the recipient understands or can offer support in the aftermath. Overall, these results indicate that family members may play a valuable role when designing ways to facilitate personal growth in the aftermath of an attempt experience.

Another key observation is that social support from family and friends can influence PTG. Indeed, the pattern of results suggests that support not only had direct effects on PTG but may also be a mechanism whereby disclosure affects PTG. Therefore, our findings align well with an abundance of work showing the benefit of social support to health, mental health, and suicide prevention [39,40,41]. They are also consistent with prior work linking social support to PTG among people who have lived through suicide loss [11,14], as well as other experiences that can be both stigmatized and traumatizing (e.g., people living with HIV/AIDS; [38]). Additionally, our results fit well with Tedeschi and Calhoun’s cognitive model of PTG [42], which posits that internal, cognitive changes occur and allow for restructuring of one’s life narrative. The disclosure of that narrative may lead to social support, which, in turn, promotes PTG. Neimeyer, Klass, and Dennis [43] support this approach in the social constructivist approach to experiences of grief. Grief may be viewed as a broad construct encompassing the experiences of both losing a loved one or nearly losing one’s own life. According to this social construction approach, narratives are created within an individual as well as between individuals. These exchanges may both promote growth as well as social support, which is often viewed as a facilitator of growth processes [14]. The good news is that sources of social support have increased for attempt survivors over the past decade [44,45]. The rise of the lived experience movement has created more in-person (e.g., support groups; [46]) and online spaces (e.g., livethroughthis.org) where attempt survivors are empowered to share experiences that might otherwise be concealed. With that said, social support from different sources may confer different levels of PTG, and the present findings call for more research in this area. If our findings can be replicated via more rigorous research designs, the field will need new strategies that specifically cultivate supportive relationships with friends and family members. 

An interesting finding is that disclosure to healthcare providers was linked with PTG only after controlling for quality of family and friend support. It could be that quality of support acted as a suppressor variable in our model, in that the correlation between quality of support and PTG suppressed enough irrelevant variance to display healthcare disclosure as a statistical predictor. However, if our findings are representative for suicide attempt survivors, it is admittedly surprising to see that disclosing more to healthcare providers was linked with lower levels of PTG: the opposite direction of other disclosure predictors. In addition to positive benefits (e.g., receiving help and support), individuals with depression and past suicidality have reported various negative consequences as a result of disclosing suicidality in a healthcare setting: losing control over their own treatment, involuntary hospitalization, disempowerment, as well as shame and judgment [47]. Interviews with physicians have shown that some healthcare providers are able to assess for risk but feel unsure about how to proceed [48] which, at minimum, likely affects their ability to appear empathetic and supportive and, at worst, leads a suicidal individual to feel even more hopeless following an attempt. In reality, there are likely a broad range of both disclosure and reaction experiences with healthcare providers, as one individual could even have both good and bad experiences across multiple providers in their lifetime. Nonetheless, this finding suggests the need for more studies that explore deterrents for disclosure to healthcare providers and how to remedy experiences that were originally stigmatizing.

Findings from this study contradict our previous work regarding the importance of a confidant. Our previous research showed that disclosing to a confidant and receiving a helpful reaction leads to less severe depressive symptoms [49], less perceptions of burdensomeness, and greater feelings of belongingness [19]. Similarly, interviews with attempt survivors showed that confidants were repeatedly utilized during times of crisis and that individuals more commonly disclosed if they could identify at least one confidant [50]. However, the present study indicated disclosure to a confidant (i.e., one person to whom you tell most things) was not a statistical predictor of posttraumatic growth or depreciation. One explanation is that there are limits to what one confidant can do in the face of a traumatic experience. It could be that a confidant is especially important during a crisis moment when immediate support is needed; nonetheless, disclosure about one’s experience is needed on a broader scale in order to integrate the experience in a way that generates posttraumatic growth. A caveat to this explanation is that confidant disclosure was linked with neither growth nor depreciation in the present study, which would suggest that it also neither helps nor harms in this context. An alternative explanation could be that there was not enough variation in the depth of disclosure to confidants to account for changes in posttraumatic growth outcomes. It will be important for more research to continue exploring the role of confidants both in an immediate crisis and in the short- and long-term process of recovery post-attempt to elucidate this idea further.

A final observation that deserves consideration is the relationship between time since one’s suicide attempt, social support, and posttraumatic changes. To summarize, we found that time since attempt was not implicated in PTG but was implicated in posttraumatic depreciation. In particular, more time since attempt was associated with less posttraumatic depreciation but only *before* controlling for social support. The absence of a relationship between PTG and time since attempt contrasts with some prior work on the subject; nonetheless, those empirical findings are mixed and were conducted with loss survivors [51]. Still, the same logic behind the link between time and PTG (e.g., “time heals wounds” [51]) could also be applied to explaining the link between time and posttraumatic depreciation. Notably, our pattern of results suggests that time might *appear* to heal, wounds but, in fact, a more active mechanism in social support is responsible for the effect on depreciation. This explanation is somewhat consistent with research suggesting that a problem-focused coping style, which can include seeking support from others, may facilitate personal growth among suicide loss survivors regardless of the amount of time that has passed since the suicide loss [11].

### 4.1. Clinical Implications

Clinicians and practitioners working with suicide attempt survivors should consider the implications of these findings. First, it is important to have specific conversations about disclosure that focus on the people to whom the individual wishes to disclose their experience and how much of it they hope to share. These conversations should also consider the potential reactions of recipients, and individuals should develop a plan for how to cope with unhelpful or hurtful responses. Although there might be some net benefits from disclosing regardless of response—as this study suggests—there could also be negative immediate consequences that warrant attention; developing a plan in advance of those responses could help to mitigate their negative effects. Similarly, clinicians should invite family members or friends (i.e., any recipients of suicide-related disclosure) to sessions to process previous disclosure conversations (e.g., how the conversation was experienced by the survivor and disclosure recipient; reframing reactions that were unintentionally hurtful) and prepare them for future disclosure conversations (e.g., education about supportive responses). If these recipients are unable to attend sessions in person, conducting a phone call could also be beneficial. Finally, clinicians should also remain aware of the potential negative effects of disclosure to healthcare providers. Mental health professionals should be prepared to explore and process past negative disclosure experiences with providers to rebuild trust in the therapeutic alliance and broader healthcare system.

### 4.2. Limitations

The findings of the present study should be considered alongside the study’s limitations. This study was conducted cross-sectionally using a convenience sample of volunteers. While our recruitment strategy might be appropriate given that there is no sampling frame for attempt survivors (i.e., a hidden population [52]), it yielded a sample that primarily identified as non-Hispanic White and female, thereby limiting our ability to generalize the findings and implications. Our sample was also recruited from listservs or accounts moderated by supportive organizations, which may not be reflective of all attempt survivor experiences; individuals in our sample may be more likely to have stronger social ties. Homogeneous samples are very common when using convenience sampling via online invitations, but there are strategies for future studies to increase diversity in their samples (e.g., with respect to sex, gender identity, or ethnicity). For instance, respondent-driven sampling can be a useful approach for addressing challenges pertaining to generalizability in work with hidden populations [52]. Alternative recruitment strategies may also be impactful in reaching a more diverse group of attempt survivors, such as tailoring invitations for and posting invitations in forums designed for diverse populations. Recruitment messages should emphasize why the field needs to hear from each group and how their participation would help others from similar backgrounds. Future research could also benefit from community-based participatory research that involves diverse groups in all stages of a study. Our study also did not account for varying experiences among people within the same group. We measured disclosure to family, friends, and healthcare providers *in general*, which did not allow us to assess whether there were large variations within that general score. For example, an individual might have disclosed their whole story to one parent yet limited information to another. Further, our study did not measure current suicidal ideation. Similarly, although we measured posttraumatic depreciation, our study mostly focused on recovery-related variables, which impacts our ability to speak to all possible outcomes. Finally, the cross-sectional nature of our research design limits our confidence in concluding that posttraumatic growth and depreciation were truly influenced by our independent and/or control variables. Follow-up work should consider study designs that are better able to establish causality, such as longitudinal approaches to examine the relationship between disclosure and posttraumatic growth/depreciation over time and/or intervention approaches that evaluate the effects of disclosure programming on posttraumatic growth/depreciation.

## 5. Conclusions

Given the high number of individuals who survive a suicide attempt each year, it is important to understand factors that predict both posttraumatic growth and depreciation following the event. Findings from the current study demonstrate that suicide-related disclosure to family members and friends statistically predict increases in posttraumatic disclosure even when controlling for time since attempt and the quality of family and friend support. Future research can build on these findings by exploring differences between disclosure to family versus friends and the impact it has on posttraumatic outcomes. More information is needed to understand how it relates to depreciation following a suicide attempt and how to lessen this negative effect.

## Figures and Tables

**Table 1 ijerph-16-03815-t001:** Means, standard deviations, and intercorrelations for independent and dependent variables.

Variable	*M*	*SD*	1	2	3	4	5	6	7	8
1. Confidant disclosure	2.70	0.83	—							
2. Family disclosure	1.72	0.71	0.35 ***	—						
3. Friend disclosure	2.03	0.75	0.48 ***	0.34 ***	—					
4. Healthcare disclosure	2.26	0.91	0.50 ***	0.31 ***	0.34 ***	—				
5. Posttraumatic growth	1.94	1.27	0.26 **	0.33 ***	0.36 ***	0.03	—			
6. Posttraumatic depreciation	1.16	1.03	−0.08	−0.09	−0.05	−0.11	−0.29 ***	—		
7. Time since attempt	―	―	0.04	0.11	−0.04	−0.15	0.15	−0.26 **	—	
8. Family support	3.88	1.79	0.21 **	0.42 ***	0.16	0.11	0.39 ***	−0.41 ***	0.21 *	—
9. Friend support	4.89	1.57	0.19 *	0.12	0.40 ***	0.20*	0.44 ***	−0.39 ***	0.06	0.31 ***

Note. * *p* < 0.05. ** *p* < 0.01. *** *p* < 0.001.

**Table 2 ijerph-16-03815-t002:** Multivariate Effects from Multivariate Analyses of Variance (MANOVA).

**Multivariate Effects from MANOVA for Posttraumatic Growth and Depreciation Measures when Controlling for Time Since Attempt**			
Predictor	F(4,155)	*p*	η^2^
Confidant disclosure	1.63	0.137	0.11
Family disclosure	2.56	0.019	0.16
Friend disclosure	3.59	0.002	0.21
Healthcare disclosure	1.22	0.299	0.08
*Covariate:*			
Time since attempt	1.94	0.073	0.13
**Multivariate Effects from MANOVA for Posttraumatic Growth and Depreciation Measures when Controlling for Time Since Attempt, Perceived Family, and Friend Support**			
Predictor	F(4,155)	*p*	η^2^
Confidant disclosure	1.60	0.144	0.11
Family disclosure	2.97	0.008	0.19
Friend disclosure	3.32	0.003	0.20
Healthcare disclosure	1.28	0.267	0.09
*Covariates:*			
Time since attempt	1.30	0.260	0.09
Family support	5.01	<0.001	0.28
Friend support	3.54	0.002	0.21

Note. Multivariate F ratios were generated from Pillai’s statistic. Effect sizes are partial eta-squared (η^2^).

**Table 3 ijerph-16-03815-t003:** Univariate Effects from Multivariate Analyses of Variance (MANOVA).

**Univariate Effects from Multivariate Analyses of Variance (MANOVA) for Posttraumatic Growth and Depreciation Measures**																						
	**Overall Growth**			**Posttraumatic Growth Subscales**															**Overall Depreciation**			
				**Relating to Others**			**New Possibilities**			**Personal Strengths**			**Spiritual Change**			**Appreciation of Life**						
Predictor	F ^a^	*p*	η^2^	F ^a^	*p*	η^2^	F ^a^	*p*	η^2^	F ^a^	*p*	η^2^	F ^a^	*p*	η^2^	F ^a^	*p*	η^2^	F ^a^	*p*		η^2^
Confidant disclosure	0.41	0.523	<0.01	1.42	0.236	0.01	0.91	0.342	0.01	0.03	0.858	<0.01	1.95	0.166	0.02	1.39	0.241	0.01	0.01	0.953		<0.01
Family disclosure	11.35	0.001	0.10	9.63	0.002	0.09	11.87	0.001	0.11	9.54	0.003	0.09	0.44	0.511	0.01	10.47	0.002	0.10	0.09	0.763		<0.01
Friend disclosure	7.21	0.009	0.07	8.62	0.004	0.08	2.93	0.090	0.03	4.82	0.030	0.05	10.82	0.001	0.10	1.34	0.251	0.01	0.05	0.830		<0.01
Healthcare disclosure	2.33	0.130	0.02	1.47	0.228	0.02	2.54	0.114	0.03	0.65	0.423	0.01	1.07	0.304	0.01	4.02	0.048	0.04	1.66	0.200		0.02
*Covariate:*																						
Time since attempt ^b^	2.44	0.121	0.02	0.50	0.482	0.01	4.51	0.036	0.04	1.76	0.187	0.02	2.02	0.158	0.02	2.03	0.158	0.02	6.63	0.012		0.06
**Univariate Effects from MANOVA when Controlling for Perceived Family and Friend Support**																						
	**Overall Growth**			**Posttraumatic Growth Subscales**																**Overall Depreciation**		
				**Relating to Others**			**New Possibilities**			**Personal Strengths**			**Spiritual Change**			**Appreciation of Life**						
Predictor	F ^a^	*p*	η^2^	F ^a^	*p*	η^2^	F ^a^	*p*	η^2^	F ^a^	*p*	η^2^	F ^a^	*p*	η^2^	F ^a^	*p*	η^2^		F ^a^	*p*	η^2^
Confidant disclosure	0.48	0.488	0.01	1.75	0.189	0.02	1.02	0.315	0.01	00.04	0.848	<0.01	2.15	0.146	0.02	1.62	0.203	0.02		0.03	0.873	<0.01
Family disclosure	6.36	0.013	0.06	3.96	0.049	0.04	6.73	0.011	0.07	6.95	0.010	0.07	0.01	0.977	0.01	7.42	0.008	0.07		1.54	0.217	0.02
Friend disclosure	2.18	0.143	0.02	3.38	0.069	0.03	0.39	0.536	0.01	0.92	0.339	0.01	6.88	0.010	0.07	0.01	0.979	0.01		1.74	0.191	0.02
Healthcare disclosure	4.39	0.039	0.04	2.90	0.092	0.03	4.07	0.049	0.04	1.63	0.205	0.02	1.36	0.246	0.01	6.31	0.014	0.06		1.69	0.196	0.02
*Covariates:*																						
Time since attempt ^b^	0.31	0.582	<0.01	0.26	0.614	0.01	1.66	0.201	0.02	0.28	0.596	<0.01	0.69	0.409	0.01	0.44	0.508	0.01		2.60	0.409	0.03
Family support	6.04	0.016	0.06	10.35	0.002	0.10	4.41	0.038	0.04	1.26	0.265	0.01	2.55	0.113	0.03	1.36	0.246	0.01		14.78	<0.001	0.13
Friend support	18.00	<0.001	0.16	16.18	<0.001	0.14	11.51	0.001	0.11	16.30	<0.001	0.14	2.75	0.100	0.03	13.54	<0.001	0.12		8.70	0.004	0.08

*Note.* Effect sizes are partial eta-squared (*η*^2^*).*
^a^ Univariate *df* = 1, 158. ^b^ Dichotomous categories: 1 = Attempt within past 5 years, 2 = Most recent attempt over 5 years ago.

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
