# Peer review of "Recipients of Suicide-Related Disclosure: The Link between Disclosure and Posttraumatic Growth for Suicide Attempt Survivors"

_ijerph, 2019, doi:10.3390/ijerph16203815_

Round 1

Reviewer 1 Report

This paper presents an exploration of posttraumatic growth (PTG) following disclosure for suicide attempt survivors, with the main finding that suicide-related disclosure to friends and family members predicts increases in PTG even when time since attempt and quality of support are controlled for. This is an interesting paper based on a relatively small but important sample of self-selected suicide attempt survivors who completed a survey based on existing scales of measurement.

The rationale for this study is sound, in wanting to understand the factors that predict PTG and depreciation following a suicide attempt, though I would like to see recommendations extrapolated from these findings. The authors suggest that research can build on this - they themselves are adding to the literature with various papers exploring disclosure of suicide attempts and ideation. It would be helpful for the reader to have a little more guidance on what this might mean in practice – e.g. perhaps clinicians and others who work with people who have suicidal ideation and/or have attempted suicide might encourage their clients to disclose to friends/family where they feel safe and able to. You might highlight more clearly your conclusion that the findings fit with a problem-focused coping-style, including seeking support. Importantly, clinicians need to be aware of the potential negative effects of disclosing to healthcare providers.

I found the text dense in places and wonder if this may hinder the reader from focusing on the key findings that you are keen to highlight. My personal preference in your results section would be to report numbers in the tables only, and remove these from the text. This would allow the reader to more easily follow your narrative of the effects found on PTG and depreciation, with reference to the numbers in the tables should they wish.

Reviewer 2 Report

Thank you for an opportunity to review the manuscript. As the authors have pointed out the area covered by the study is relatively new and unexplored, although. e.g., Heckler's book 'Waking Up, Alive' (1994) flagged the role of PTG, growth and recovery in the aftermath of suicide attempt.

The major problems with the study are a) the design (no control group), b) lack of exploration of heterogeneity in the study sample (e.g., how many suicide attempts, treatment/support received in the aftermath od a suicide attempt, current suicidal ideation, etc), and c) selective inclusion of recovery-related  variables in the study.

There is a risk that the study sample was highly selective, e.g., online volunteers, well functioning individuals in recovery (how to define recovery?), thus significantly limiting generalisibility of the results and conclusions.

I will be looking forward to reading authors' responses to these points.

Reviewer 3 Report

Thankyou for inviting me to review this manuscript about posttraumatic growth preceeding a suicide attempt.

I think this adds an important contribution to the field, and this is to me an interesting and new perspective.

I have only made some minor comments and suggestions to the authors: 

Terminology

Maybe the term nonfatal suicide attempt can be replaced with suicide attempt and used consequently throughout the paper? Every suicide attempt is a nonfatal suicide isn’t it? 

Table 1 in Notes, is the last p-value missing a star?

Note. *p < .05. **p < .01. **p < .001

I think that the time since suicide attempt variable is important, but miss more reflections about why and how this will influence the PTG?

In the methods section page 3, line 90-100, I think maybe the paper would improve by describing the wording of the invitation to provide a more thorough description of the included sample.

Page 3, line 121; Insert the reference for the CDDS scale.

Page 4, line 181: in Europe the term: Race is not political correct to use.

Please consider the importance of this variable in the study.

Discussion page 11, line 250: If you start with a resyme of the most important findings it will improve the readability, instead of waiting until line 254. 

Limitations page 13 line 358, explain what you mean with diverse populations?

I also think that the paper would improve by adding some facts about how many of those attempting suicide struggle for a long period after the attempt, and that this paper probably did not include those who are not healthy or have enough resources to seek help in those places where this sample was recruited.

Round 2

Reviewer 2 Report

The authors have addressed the issue of study design.

I am still concerned about the b) lack of exploration of heterogeneity in the study sample (e.g., how many suicide attempts, treatment/support received in the aftermath od a suicide attempt, current suicidal ideation, etc), and c) selective inclusion of recovery-related variables in the study.

Please, respond to these two queries and/or acknowledge these two points flagged here as limitations. 
